# A Two-Level Rolling Optimization Model for Real-time Adaptive Signal Control

**Zhihong Yao [1,2,3,*]** 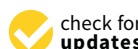**, Yibing Wang [4], Wei Xiao [2], Bin Zhao [2] and Bo Peng [5]**

[1]  Chongqing Key Laboration of Traffic and Transportation, Chongqing Jiaotong University, Chongqing 400074, China

[2]  School of Transportation and Logistics, Southwest Jiaotong University, Chengdu 611756, China; emily.wx@my.swjtu.edu.cn (W.X.); binzhao@my.swjtu.edu.cn (B.Z.)

[3]  Department of Civil and Environmental Engineering, University of Wisconsin-Madison, Madison, WI 53706, USA

[4]  Department of Architecture Engineering, Yantai Vocational College, Yantai 264670, China; ybwang30@gmail.com

[5]  School of Traffic and Transportation, Chongqing Jiaotong University, Chongqing 400074, China; bpeng.cqjtu@gmail.com

*  Correspondence: zhyao@my.swjtu.edu.cn; Tel.: +86-155-2825-0279

**Abstract:** Recently, dynamic traffic flow prediction models have increasingly been developed in a connected vehicle environment, which will be conducive to the development of more advanced traffic signal control systems. This paper proposes a rolling optimization model for real-time adaptive signal control based on a dynamic traffic flow model. The proposed method consists of two levels, i.e., barrier group and phase. The upper layer optimizes the length of the barrier group based on dynamic programming. The lower level optimizes the signal phase lengths with the objective of minimizing vehicle delay. Then, to capture the dynamic traffic flow, a rolling strategy was developed based on a real-time traffic flow prediction model. Finally, the proposed method was compared to the Controlled Optimization of Phases (COP) algorithm in a simulation experiment. The results showed that the average vehicle delay was significantly reduced, by as much as 17.95%, using the proposed method.

**Keywords:** adaptive signal control; dynamic programming; integer linear programming; rolling strategy; vehicle delays

## 1. Introduction

With the development of the social economy, traffic congestion has become one of the most significant problems in many cities. Traffic signal control is a critical form of traffic control and management to reduce urban traffic congestion. Traffic signal control theory has been established for over 60 years, starting from the pioneering work of Webster [1]. Since then, research and development in traffic signal control has largely fallen into three types of control strategies: fixed-time control, actuated control, and responsive control.

Fixed-time control is based on historic traffic data and assumes traffic demand is constant. Actuated control uses preset rules to adapt traffic flow based on detected traffic data (mainly vehicles passing/existing). Responsive control optimizes the signal timing plans based on real-time detected traffic data and improves the usage of intersection capacity [2–4]. There are a few widely used responsive traffic control systems in the world [5–7]: SCATS [8] was developed in Australia, SCOOT [9] was developed in Britain, RODYN [10] and CRONOS [11] were developed in France, UTOPIA [12] was developed in Italy, OPAC [13] and RHODES [14,15] were developed in the USA.

The optimization algorithm, which can generate the optimal signal timing plans based on a given objective, is regarded as an indispensable part of adaptive control systems. At present, the optimization algorithm in adaptive control systems can be divided into the following categories [16]: dynamic programming [14,17,18], genetic algorithms [19,20], neural networks [21,22], and fuzzy logic control [23,24]. Because of the fast calculation speed, the dynamic programming algorithm is widely used in adaptive control systems such as PRODYN [10], UTOPIA [12], OPAC [13] and RHODES [14,15]. In the RHODES system, the optimization algorithm is the dynamic programming algorithm named, Controlled Optimization of Phases (COP) [14]. In 2015, Feng et al. [17] proposed a dynamic programming signal timing optimization algorithm based on connected vehicle data. However, these algorithms do not consider the time-varying characteristics of the traffic flow during the optimization horizon. The longer the optimization horizon is, the worse the prediction effect is. Therefore, signal timing plans based on these methods are often unsatisfactory.

With the development of connected vehicle technology, the time granularity of prediction data is becoming smaller and smaller [25]. Therefore, the use of small granularity predicted data to optimize the signal timing plans has been a popular research topic in the last couple of years. To address this issue, a two-level rolling optimization model for real-time adaptive signal control based on a dynamic traffic flow model was developed. The proposed method consists of two levels, i.e., barrier group and phase. The upper layer optimizes the length of the barrier group based on dynamic programming. The lower level optimizes the signal phase lengths with the objective of minimizing vehicle delay. Then, to capture the dynamic traffic flow, a rolling strategy was developed based on a real-time traffic flow prediction model. Finally, the proposed method was compared with the Controlled Optimization of Phases (COP) algorithm in a simulation experiment.

The remainder of this paper is structured as follows. The traffic signal timing optimization model and algorithm are described in Section 2. In Section 3, a case study and discussion about the proposed method are presented. Conclusions and future work are discussed in Section 4.

## 2. Traffic Signal Timing Optimization Algorithm

The signal timing optimization algorithm optimizes signal phase durations based on a dynamic traffic flow prediction model. The optimization method consists of two levels of optimization in this paper. At the upper layer, a dynamic programming (DP) is applied to each barrier group, with each barrier group between two barriers defined as a phase group. Based on the above definition, a standard National Electrical Manufacturers Association (NEMA) ring barrier controller structure is shown in Figure 1. The figure illustrates a phase sequence with left-turn movements leading the opposing through movements on both the major and minor streets. The diagram shows phases 1 and 5 ending at different times. The subsequent phase (phases 2 and 6 respectively) may begin once the previous phase has used its time. Once the barrier is crossed, phases 3 and 7 operate followed by phases 4 and 8. The cycle ends with the completion of phases 4 and 8. The calculation of the performance function of the upper level is passed to the lower level. The lower level (individual phase) optimization is formulated as an integer linear programming problem. In this study, the objective is to minimize the total vehicle delay and the sequence of barrier groups is assumed to be fixed. Time is discretized to 1 sec intervals, and the optimization is performed on a predetermined planning horizon, e.g., 80 s. Therefore, the problem is to find an optimal length for each phase to minimize vehicle delay. In addition, some constraints should be considered, such as the minimal and maximal green time in each phase. The two-level optimization model will be discussed in the next section.

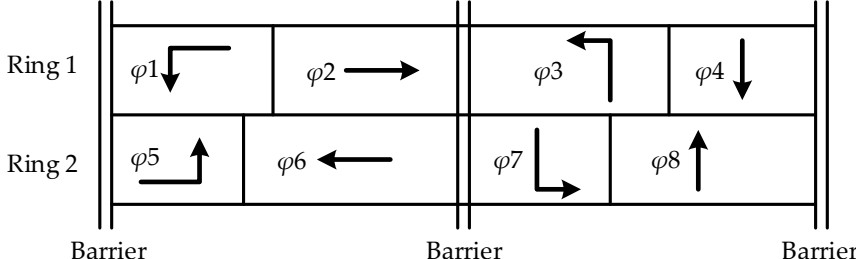

**Figure 1.** The standard NEMA dual-ring controller diagram.

## 2.1. Dynamic Programming Algorithm

The upper optimization is a DP problem, and a forward and a backward recursion are used to solve this. Figure 2 shows the relation of some variables, and Table 1 lists the notation of parameters and variables used in the DP algorithm.

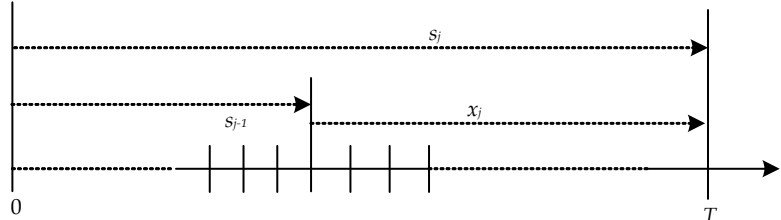

**Figure 2.** The relationship between the states, decisions, and the total number of discrete time-steps.

**Table 1.** Notation of key parameters and variables used in the dynamic programing (DP) algorithm.

| Variable | Description |
|---|---|
| $p$ | Phase index in each ring and barrier group, $p = 1, 2$. |
| $r$ | Ring index in each barrier group, $r = 1, 2$. |
| $j$ | Index of barrier groups/stages. |
| $J$ | Last stage calculated by the DP before stopping. |
| $x_j$ | Decision variable denoting the length of barrier group $j$. |
| $s_j$ | State variable denoting the total number of time steps from the start time to barrier group $j$. |
| $S_j$ | A set of state variable $s_j$. |
| $T$ | The total number of discrete time steps in the planning horizon, seconds. |
| $X_j(s_j)$ | A set of feasible control decisions, given barrier group state $s_j$. |
| $f(s_j, x_j)$ | Performance measure (objective function) at stage $j$, given barrier group state $s_j$ and control variable $x_j$. |
| $v_j(s_j)$ | Value function (cumulative value of prior performance measures), given state variable $s_j$. |
| $R_{r,p}$ | Phase change interval which is the total of the yellow change and red clearance times of phase $p$ in ring $r$. |
| $G_{r,p}^{min}$ | Minimum green time of phase $p$ in ring $r$. |
| $G_{r,p}^{max}$ | Maximum green time of phase $p$ in ring $r$. |
| $X_j^{min}$ | Minimum possible barrier group length of stage $j$. |
| $X_j^{max}$ | Maximum possible barrier group length of stage $j$. |
| $g_{r,p}$ | Green time of phase $p$ in ring $r$. |

In this paper, a forward and a backward recursion were used to solve the DP problem. The forward recursion calculates the performance measure (objective function) based on the decision and state variables and records the optimal value function for each stage. The backward recursion retrieves the optimal policy starting from the final stage and working backward. The details of the forward and backward recursion are described below.

The forward recursion is based on the allocation of time to each barrier group as stages in the DP. Considering each barrier group as a stage, the algorithm plans as many stages as necessary to obtain the optimal solution. The ring and phase within one barrier group are defined in Figure 1. The phases in each barrier group are divided into two rings, and $r$ represents the ring index and $p$ represents the

phase index within the ring. Due to the variability of traffic demand, the algorithm will not produce a fixed cycle length.

The minimum and maximum allowable barrier group lengths are calculated according to the signal timing parameters as shown in Equations (1) and (2). The parameters include the minimum green, maximum green, yellow change and red clearance times of each phase.

$$X_j^{min} = \max\left\{G_{1,1}^{min} + R_{1,1} + G_{1,2}^{min} + R_{1,2}, G_{2,1}^{min} + R_{2,1} + G_{2,2}^{min} + R_{2,2}\right\}, \tag{1}$$

$$X_j^{max} = \min\left\{G_{1,1}^{max} + R_{1,1} + G_{1,2}^{max} + R_{1,2}, G_{2,1}^{max} + R_{2,1} + G_{2,2}^{max} + R_{2,2}\right\}. \tag{2}$$

Then, given the state $j$ and the calculated minimum and maximum time for that barrier group, and the total discrete time-steps, the set of state variables is determined by Equation (3).

$$S_j = \left\{\min\left\{\sum_{k=1}^{j} X_k^{min}, T\right\}, \cdots, \min\left\{\sum_{k=1}^{j} X_k^{max}, T\right\}\right\} \tag{3}$$

Given the state variable $s_j$ and the calculated minimum and maximum time for the barrier group, the set of feasible decision variables are determined by Equation (4).

$$X_j(s_j) = \left\{\max\left\{X_j^{min}, s_j - \sum_{k=1}^{j-1} X_k^{max}\right\}, \cdots, \min\left\{X_j^{max}, s_j - \sum_{k=1}^{j-1} X_k^{min}\right\}\right\} \tag{4}$$

After determining the equations of $S_j$ and $X_j(s_j)$, DP is used to search for the best signal timing plan. The forward recursion is described as follows.

### 2.1.1. Forward Recursion

Step 1:　Set $j = 1$, $s_{j-1} = 0$ and $v_j(0) = 0$.

Step 2:　Calculate $S_j$.

Step 3:　For $s_j$ in $S_j$ {

　　　　Calculate $X_j(s_j)$.

$$v_j(s_j) = \text{Min}_{x_j}\left\{f_j(s_j, x_j) + v_{j-1}(s_{j-1}) \,|\, x_j \in X_j(s_j)\right\}$$

　　　record $x_j^*(s_j)$ as the optimal solution in Step 2.

　　　}.

Step 4:　If $(\sum_{k=1}^{(j+1)} X_k^{min} \leq T)$, $j = j + 1$, go to Step 2.

　　　　Else STOP.

For each barrier group, DP calculates the optimal decision $x_j^*(s_j)$ for each state variable $s_j$. The performance measure (objective function) $f_j(s_j, x_j)$ used to determine the state variable is passed to the lower optimization level with the constraint of control variable $x_j$. The stopping criteria will be met if the sum of the minimum time length in all barrier groups is larger than $T$. The justification of the stopping criterion is different from that in the COP algorithm [14], which does not consider the constraint of the maximum green time of a phase group. In addition, considering that pedestrians need to cross the street, barrier groups are not allowed to be skipped in this study.

After all decisions are made for all barrier groups, the optimal decision $x_j^*(s_j)$ of each barrier group can be retrieved in the backward recursion as follows.

### 2.1.2. Backward Recursion

Step 1:　Set $j = J, s_j^* = T$.

Step 2:　For $j = J, J - 1, \cdots, 1\{$

　　　　　Read $x_j^*\left(s_j^*\right)$ from the table computed in forward recursion.

　　　　　If $(j > 1), s_{j-1}^* = s_j^* - x_j^*\left(s_j^*\right)$.

　　}

The optimal plan is retrieved from barrier group $J$ since this barrier group denotes the minimum performance measure $v_J^*(T)$, such as the minimum delays or stops.

### 2.2. Integer Linear Programming

In Step 3 of the forward recursion, $f_j(s_j, x_j)$, the optimal performance measure (objective function) at stage $j$, given barrier group state $s_j$ and control $x_j$, needs to be calculated. The value of $f_j(s_j, x_j)$ depends on the green duration of the $j$th barrier group. In this study, the vehicle delays can be considered as the objective function. Then, the lower level integer linear programming is formulated in Equations (5)–(10). To solve the integer linear programming problem, the optimal phase duration is enumerated to find the minimum delay combination for the given $x_j$. The arrival flow of each phase at each time step ($A_{r,p}(t)$) comes from a predicted arrival table, which is a two-dimensional matrix with time and phase respectively. The value in each cell is the number of vehicles that will arrive at the stop bar after time interval $t$ requesting phase $p$ in ring $r$ and is the result of the traffic flow prediction model [26].

Firstly, the cumulative delay can be calculated by using the IQA method [27]. Given $x_j$ is the length of the barrier group, the lower level problem solves one of the following optimization problems. The objective function as shown in Equation (5).

$$\min \sum_{r=1}^{2} \sum_{p=1}^{2} d\left(g_{r,p}, R_{r,p}\right), \tag{5}$$

where,

$$d\left(g_{r,p}, R_{r,p}\right) = \sum_{t=s_{j-1}+1}^{s_{j-1}+x_j} l_{r,p}(t), r = 1, 2; p = 1, 2, \tag{6}$$

where $d\left(g_{r,p}, R_{r,p}\right)$ is the total delay in the given $g_{r,p}$ and $R_{r,p}$; $l_{r,p}(t)$ denotes the queue length of phase $p$ in ring $r$ at time interval $t$.

The queue length at time interval $t$ depends on the queue length of time interval $t - 1$, the arrival and departure vehicles during time interval $t$, which follows the basic flow conservation relationship as shown in Equation (7).

$$l_{r,p}(t) = l_{r,p}(t - 1) + A_{r,p}(t) - D_{r,p}(t), r = 1, 2; p = 1, 2, \tag{7}$$

where $A_{r,p}(t)$ and $D_{r,p}(t)$ denote the number of arrival and departure vehicles of phase $p$ in ring $r$ at time interval $t$, respectively.

The analysis shows that the departure of vehicles at time interval $t$ is related to the initial queue vehicles at time interval $t$, traffic signal state, and the saturation flow rate. This relationship is shown in Equation (8).

$$\begin{aligned} D_{r,1}(t) &= \min(S_{r,1}, l_{r,1}(t - 1) + A_{r,1}(t)), \text{if } s_{j-1} + 1 \le t \le s_{j-1} + g_{r,1}, r = 1, 2. \\ D_{r,2}(t) &= \min(S_{r,2}, l_{r,2}(t - 1) + A_{r,2}(t)), \text{if } s_{j-1} + g_{r,1} + R_{r,1} < t \le s_{j-1} + g_{r,1} + R_{r,1} + g_{r,2}, r = 1, 2. \\ D_{r,p}(t) &= 0, \text{if } s_{j-1} + g_{r,1} + R_{r,1} + g_{r,2} < t \le s_{j-1} + x_j, r = 1, 2; p = 1, 2. \end{aligned} \tag{8}$$

where $S_{r,p}$ denotes the saturation flow rate of phase $p$ in ring $r$, veh/s.

Then, considering the duration of two barrier groups in each ring, these should be equal to the current decision variable $x_j$ Equation (9). In addition, the duration of each phase is bounded by a lower limit and an upper limit which is shown in Equation (10).

$$g_{r,1} + R_{r,1} + g_{r,2} + R_{r,2} = x_j, r = 1, 2. \tag{9}$$

$$G_{r,p}^{\min} \leq g_{r,p} \leq G_{r,p}^{\max}, r = 1, 2; p = 1, 2. \tag{10}$$

### 2.3. Rolling Strategy

To avoid the effect caused by predicted vehicle arrival errors and to use the newly collected data, the rolling strategy based on DP algorithm was proposed. The algorithm is solved at the end of each time step based on a rolling strategy, as shown in Figure 3.

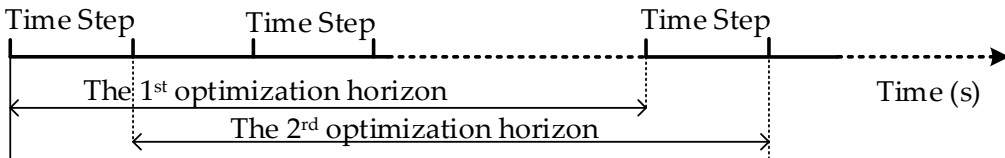

**Figure 3.** Diagram of rolling horizon optimization strategy.

From Figure 3, if each optimization horizon includes n time steps, which is related to the length of the prediction horizon, then, the first optimized time horizon will be 1 to n time steps. After the first time step is finished, the second optimized time horizon is 2 to n + 1 time steps. By analogy, when a time step is finished, a new optimization is performed immediately based on the latest forecast data. Therefore, the length of the time step is an important parameter in the rolling strategy. The different time step lengths will be discussed in the case study.

### 3. Case Study and Discussion

As shown in Figure 4, there is an actual road network in Chengdu, China, which has a typical grid structure. Geometric data were collected in the field to reflect real conditions and were further modeled into the microscopic simulation software Vissim [28]. Of the five intersections in total, one of those intersections, which is marked 5 in Figure 4, was chosen as the testing intersection for the proposed control system. The area surrounding the intersection is necessary for traffic flow prediction model [26] and to ensure realistic traffic flows. Full-actuated control was applied to all the other intersections. As a reasonable simplification, no right-turn traffic was modeled in this study, and only straight and left-turn traffic flow were modeled.

The simulation pre-warm time was set at 900 secs, and the effective simulation time was 3600 secs. Different traffic volume levels were modeled in Vissim [28] for testing the compliance of the proposed control system with real traffic conditions. In addition, to analyze the sensitivity of the rolling time step, we use three rolling time steps: 2 secs, 4 secs, and 6 secs. The average delays of the two control methods were collected from simulation data and are plotted in Figure 5.

As shown in Figure 5, the average vehicle delay in the two methods increased with the increase of traffic volume. However, compared with the COP algorithm, the proposed control method always had a lower average vehicle delay. The benefits are mainly due to the proposed method being based on a rolling strategy, which can capture the real-time chrematistics of traffic flow. Therefore, the proposed method had a smaller vehicle delay. In addition, as shown in Figure 5, the smaller the time step of rolling optimization, the better the effect of the proposed control method. This showed that smaller rolling time steps can bring better performance.

Next, the average vehicle delay of each phase was obtained as shown in Tables 2–4.

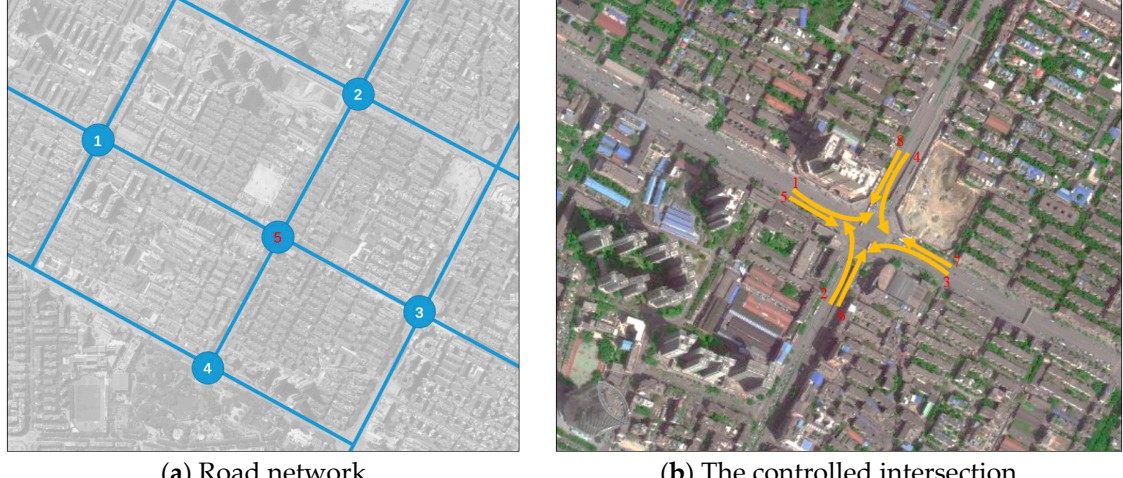

(**a**) Road network　　　　　　　　　　　　(**b**) The controlled intersection

**Figure 4.** Diagram of the simulated road network.

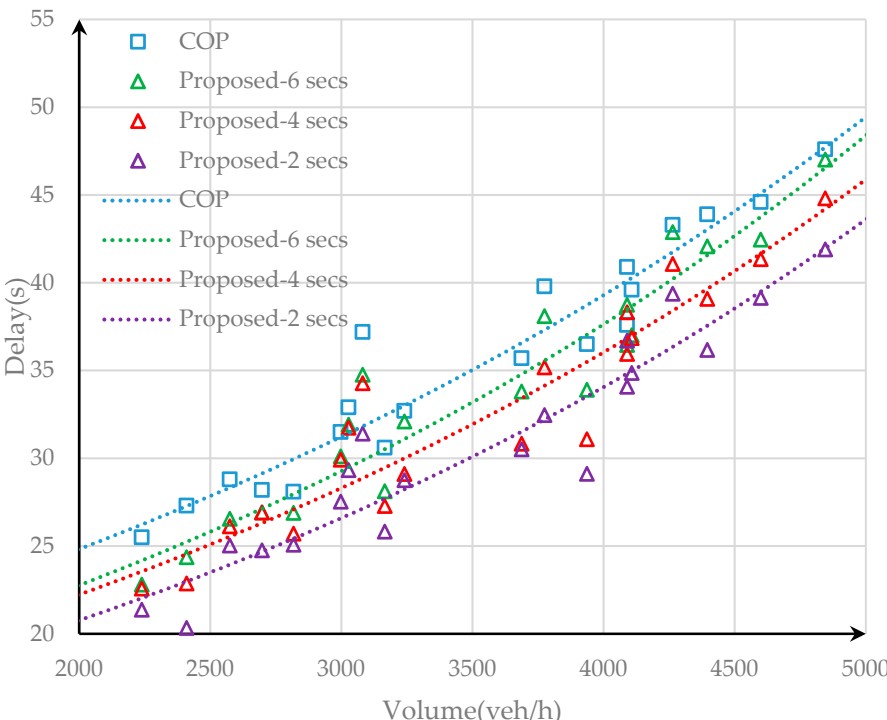

**Figure 5.** The control performance versus volume for Controlled Optimization of Phases (COP) and the proposed method.

**Table 2.** The average vehicle delay of each phase (volume level is 4500 veh/h).

| Methods | Phase 1 | Phase 2 | Phase 3 | Phase 4 | Phase 5 | Phase 6 | Phase 7 | Phase 8 | Average | Improvement |
|---|---|---|---|---|---|---|---|---|---|---|
| COP | 43.40 | 34.70 | 53.30 | 39.60 | 38.60 | 48.20 | 50.40 | 37.10 | 43.16 | NA |
| Proposed-6 secs | 41.23 | 32.03 | 51.46 | 38.77 | 36.27 | 46.12 | 50.11 | 34.61 | 41.32 | −4.26% |
| Proposed-4 secs | 38.42 | 30.68 | 50.52 | 36.46 | 36.06 | 43.77 | 48.03 | 34.09 | 39.76 | −7.89% |
| Proposed-2 secs | 36.51 | 30.54 | 48.90 | 34.07 | 35.15 | 41.92 | 45.83 | 31.72 | 38.08 | −11.78% |

**Table 3.** The average vehicle delay of each phase (volume level is 3500 veh/h).

| Methods | Phase 1 | Phase 2 | Phase 3 | Phase 4 | Phase 5 | Phase 6 | Phase 7 | Phase 8 | Average | Improvement |
|---|---|---|---|---|---|---|---|---|---|---|
| COP | 35.01 | 24.95 | 44.64 | 30.74 | 28.76 | 38.24 | 40.73 | 29.01 | 34.01 | NA |
| Proposed-6 secs | 32.52 | 24.29 | 43.83 | 27.98 | 28.15 | 37.67 | 39.38 | 26.92 | 32.59 | −4.17% |
| Proposed-4 secs | 31.98 | 23.41 | 43.11 | 27.63 | 25.29 | 36.79 | 36.40 | 26.18 | 31.35 | −7.82% |
| Proposed-2 secs | 29.46 | 23.02 | 42.10 | 25.38 | 24.73 | 34.56 | 34.56 | 24.74 | 29.82 | −12.32% |

**Table 4.** The average vehicle delay of each phase (volume level is 2500 veh/h).

| Methods | Phase 1 | Phase 2 | Phase 3 | Phase 4 | Phase 5 | Phase 6 | Phase 7 | Phase 8 | Average | Improvement |
|---|---|---|---|---|---|---|---|---|---|---|
| COP | 25.63 | 15.99 | 36.12 | 21.27 | 19.16 | 30.04 | 31.83 | 20.30 | 25.04 | NA |
| Proposed-6 secs | 22.67 | 13.87 | 33.50 | 19.00 | 18.86 | 27.99 | 29.99 | 17.74 | 22.95 | −8.35% |
| Proposed-4 secs | 20.72 | 12.07 | 30.73 | 17.84 | 18.34 | 26.89 | 29.52 | 17.01 | 21.64 | −13.60% |
| Proposed-2 secs | 19.63 | 11.79 | 28.39 | 17.35 | 17.98 | 25.23 | 28.71 | 15.29 | 20.55 | −17.95% |

As shown in Tables 2–4, the reduction of average delay is observed for all phases. For the studied intersection, the proposed control method reduced the average vehicle delay in each phase, compared with the COP algorithm. The results show that the proposed method was able to reduce both total vehicle delay of the intersection and for each phase. In addition, the reduced average vehicle delay was as much as 17.95%, 12.32%, and 11.78%, respectively, when the traffic volumes were 2500, 3500, and 4500 veh/h.

## 4. Conclusions and Future Work

### 4.1. Conclusions

Through the actual survey data and the simulation analysis, the following conclusions were reached in this study: (1) at the whole intersection level, the proposed algorithm has less delay than the COP algorithm, and the average vehicle delay is reduced by 17.95%; (2) At the intersection phase level, compared with the COP algorithm, the proposed algorithm can reduce the vehicle delay in each phase; (3) Smaller rolling time steps can bring better performance.

### 4.2. Future Work

The coordination control (with common cycle and coordinated offset) will be studied in future work. In addition, there are still some open areas that are worthy of being investigated, such as: multiple objectives optimization to achieve a more balanced signal control plan and feedback strategy to gain more robust control by using the post-event vehicle delay, queueing and turning data et al. supported by the new intelligent transportation technology.

**Author Contributions:** Conceptualization, Z.Y. and B.P.; methodology, Z.Y. and Y.W.; software, B.Z.; validation, Z.Y., W.X., and Y.W.; formal analysis, W.X. and B.Z.; writing—original draft preparation, Z.Y. and W.X.; writing—review and editing, Z.Y. and B.Z.; visualization, W.X.; funding acquisition, Y.W..

**Acknowledgments:** The research received funding support from the Open Fund Project of Chongqing Key Laboratory of Traffic & Transportation (2018TE01), the Chengdu Science and Technology Project (No. 2017-RK00-00362-ZF), the Chinese National Natural Science Fund (61703064), and the Chongqing Research Program of Basic Research and Frontier Technology(cstc2017jcyjAX0473). The authors are grateful to the two anonymous reviewers for sharing their research insights and providing helpful comments to improve the quality of the paper.

**Conflicts of Interest:** The authors declare no conflict of interest.

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
