# Peer review of "A Two-Level Rolling Optimization Model for Real-time Adaptive Signal Control"

_algorithms, doi:10.3390/a12020038_

Reviewer 1 Report

This paper aims to develop signal timing rolling optimization considering the dynamic change of traffic flow. For this, the authors proposed a DP algorithm in a high level where they find the optimal state (I believe this is the time the green light is on). Here I am confused as what the integer programming is used to compute the optimal duration of both phases (per ring?) which fulfill the optimal state. The algorithm runs in an endless loop inputting the new obtained data. Authors claim that they’re able to re reduce delays to 20% compared against the conventional COP.

General comments:

This journal aims to algorithm in general and its applications. For this reason, authors should improve the traffic signaling problem description as it might not be clear for readers as myself. This lack of information makes it hard to follow the developed algorithm. I would suggest to improve the problem definition section where you could briefly explain the ring barrier controller, the standard NEMA ring barrier and the different performance measurements, and the problem’s challenges. 

Authors should as well invest in explaining the state variable as it is not clear. Does this state variable represent the traffic signal sequence? Or does it reflect the times of each phase within the ring? I can infer from (1) and (2), that the state is the time the green light is on. 

You comment a two phases optimization: One with DP and other with integers. It is not clear why this is happening. I would think this is related to my lack of knowledge. 

There are some different DP algorithms in the literature for this problem. It would be good to place some of them in your reviews. By reading the titles, I was interested in this one: “Adaptive Dynamic Programming for Multi-intersections Traffic Signal Intelligent Control” by Li Tao, et al. It seems similar to your research.

From your results section I cannot see the advantage  the rolling. It is also hard to see the link with the granularity. I understand that optimizing in an infinite loop allows to introduce the new data, but you did not discuss the timing rolling influence in your results or conclusions. 

Other corrections/questions

Lines 15-17. Consider rewriting this sentence. It is too long. 

Lines 39-41. Do the authors mean that the data is collected via sensors, introduced into a model and use the output for the traffic signals or do the authors mean that the measured data is compared against a model? 

Line 45. Please briefly explain why the optimization algorithm is that important 

Lines 49- 50. It is not clear how the Dynamic Programing (DP) can be used as an “adaptive control systems”. In the previous sentences it was said that optimizations techniques are important for the adaptive control systems. 

60-61. This sentence seems to be incomplete. You said in order to solve the problem, “…”, “…”. Something is missing. 

95.  

Table 1. f(s,x) is called performance measure. In line 138 is called the objective function. Please define them. 

Typo in Rr,p description (There is a 3)

124. Call them equations instead of formulas. 

175. Is “t” a given point of time or is “t” a time frame (for example from 0 to 1 min or so)? It is hard for me to picture that at Time t, 3 vehicles can go out and 1 come in for example. Or is it an iteration?

184. typo. Two parentheses in eq (11)). 

Lines 188 to 194 could have mentioned before so the reader can have this in mind. This way, eq 10-12 make more sense. 

194. By rolling strategy you mean that you use your output and new data and you keep optimizing in an “infinite loop”?

220. What is Vissim?

231 – 237. Add labels to show that the axes represent delays (s). 

Author Response

Sincerest thanks for your comments. We have carefully reviewed the comments and have revised the manuscript accordingly. Your comments and those of the reviewers were highly insightful and enabled us to greatly improve the quality of our manuscript. 

Reviewer 2 Report

The paper entitled "Traffic Signal Timing Rolling Optimization Algorithm based on Dynamic Programming" provides a view on how the COP algorithm can be improved for achieving less delays at one specific intersection. The authors use the non-referenced software VISSIM by the German company PTV. Since the explanations in "CASE STUDY AND DISCUSSION" are rather simple and often not understandable, especially how the field conditions are "coded into VISSIM", I would leave out the software mentioning (since also no reference), since other microscopic traffic simulation software are also usable for this approach. @Investigation area: Are those OSM extracts? Modelling of network is not mentioned in the paper.

In general, the paper will find its readers, since short and precise, well-structrured, and, implies important references. Formulas and Variables are correct and the figures have an acceptable resolution. Nevertheless, there are several confusing things in the Abstract and especially the Introduction:

- "classical signal timing optimization algorithm" is abbreviated as "COP" (l. 15), actually the "Controlled Optimization of Phases"; the authors should change this or introduce COP in the Introduction for the first time.

- l. 33: "Traffic signal control theory has been established for over 150 years": I think this applies only for "control theory", if not a mistake, since Websters paper is from 1958.

- l. 59: "current research hot topic" - I'm really not sure to call it that way. Comparing to some topics in other domains, there is relatively few research. Surely, the authors can mention that has been more contributions in the last couple of years.

All in all, an acceptable paper.

Author Response

Sincerest thanks for your comments. We have carefully reviewed the comments and have revised the manuscript accordingly. Your comments and those of the reviewers were highly insightful and enabled us to greatly improve the quality of our manuscript. 

Round  2

Reviewer 1 Report

Dear Authors, 

Thank you very much for your replies. 

Just some last comments. 

Line 75-76: the objective is to minimise (not minimal)

Line 96: The performance measure (objective function). (Add objective function)

Line 180: Error! Reference source not found. (Delete it please)

General comment: 

I would suggest to explain the Nema ring barrier still. Just a brief paragraph saying what this ring means. How to read it. 

Thank you

Author Response

Thanks for your comments. We had revised the manuscript.

Reviewer 2 Report

All comments have been adressed. Nevertheless, there are some remaining typos:

l. 13: I would memove the "In" in "In recently" and simply start with "Recently"

l. 21: The results show significantly reduce average vehicle’s delay by the proposed method.

--> it is "reduced"

Small things that have to be revised:

l. 208: Geometric data is collected in the field for reflecting the real condition, and is further
209 modeled into a microscopic simulation soft, such as Vissim [28].

--> it is "software" not "soft", and, I would say it is modelled "with a simulations software".

Author Response

Thanks for your detail comments. We had revised it. Thank you.
